# OpenReview forum: "Using In-Context Learning to Improve Dialogue Safety"
_EMNLP/2023/Conference — EMNLP 2023 Findings_

### Official Review · Reviewer_QjSk · 2023-07-31

**Soundness:** 4

**Excitement:**

3: Ambivalent: It has merits (e.g., it reports state-of-the-art results, the idea is nice), but there are key weaknesses (e.g., it describes incremental work), and it can significantly benefit from another round of revision. However, I won't object to accepting it if my co-reviewers champion it.

**Missing References:**

A few other prompting methods that relates/could be used to improve dialogue safety
1. Si, C., Gan, Z., Yang, Z., Wang, S., Wang, J., Boyd-Graber, J.L., & Wang, L. (2022). Prompting GPT-3 To Be Reliable. ArXiv, abs/2210.09150.
2. Madaan, A., Tandon, N., Gupta, P., Hallinan, S., Gao, L., Wiegreffe, S., Alon, U., Dziri, N., Prabhumoye, S., Yang, Y., Welleck, S., Majumder, B.P., Gupta, S., Yazdanbakhsh, A., & Clark, P. (2023). Self-Refine: Iterative Refinement with Self-Feedback. ArXiv, abs/2303.17651.

**Paper Topic And Main Contributions:**

This paper focuses on reducing toxicity of LM generated responses by using prompting/in-context learning. The authors propose a retrieval-based framework that 1) retrieve safe demonstrations that are similar to the current dialogue context and 2) prompt the LM with those safe demonstrations to generate a response. The authors then evaluated their approach on three safety-related dialogue datasets, using both automatic metrics and human evaluation they showed that 1) such retrieval + prompting based method can reduce toxicity without reducing much engageness/coherence and 2) such no-training-required method performed competitively with other popular training methods (e.g. DIRECTOR) on reducing toxicity.

**Questions For The Authors:**

1. GPT-3.5/4 based automatic evaluation may favor longer responses (Yizhong, et al, 2023). The paper mentioned that having 10 demonstrations may cause generated responses to be longer. How would this affect the results in Table 2 and 3?

**Reasons To Accept:**

1. The authors presented a retrieval+prompting method that can be used to effectively reduce toxicity. Such no-training method performed competitively against other popular training-based methods, while not losing much dialogue engageness/coherence.
2. Numerous detailed experiments are presented to support their claims, and additional considerations such as the effect of model size and the order/number of safe demonstrations.
3. To the best of authors knowledge, this is the first large-scale evaluation of in-context learning for dialogue safety

**Reasons To Reject:**

1. Human evaluation is only performed for comparisons against baselines without any safety mechanisms.
2. Since the proposed toxicity reduction method mainly relies on the prompts designed/retrieved by the authors, it should also be compared against other prompting based methods, especially since steering the behavior of LM using specialized prompts is not entirely new. See missing references for some related methods.
3. Other weaknesses include experiments only involve short dialogue contexts (upto 2 turns). This is also acknowledged by the authors as limitations.

**Reproducibility:**

4: Could mostly reproduce the results, but there may be some variation because of sample variance or minor variations in their interpretation of the protocol or method.

**Reviewer Confidence:**

3: Pretty sure, but there's a chance I missed something. Although I have a good feel for this area in general, I did not carefully check the paper's details, e.g., the math, experimental design, or novelty.

---

> ### Author Rebuttal · Authors · 2023-08-29
>
> We thank Reviewer QjSk for their useful feedback and comments. We are happy they found our work provided "**numerous detailed experiments**" to "**support [our] claims.**" We are glad they found the competitive performance of our method exciting! We also highlight that the other reviewers found:
> * Our results "**encourgaging and promising**" (Reviewer kDkA).
> * Our paper "**well-written,**" "**easy to comprehend,**" and "**logically organized**" (Reviewer kDkA).
> * Our evaluation metrics "**diverse**" adding to the "**credibility**" of our results (Reviewer kDkA).
> * Our experiments "**comprehensive and robust**" (Reviewer kDkA).
> * Our paper topic "**timely**" (Reviewer yFp7).
>
> We address each of Reviewer QjSk’s comments below.
>
> > Human evaluation is only performed for comparisons against baselines without any safety mechanisms.
>
> We only perform human evaluation against a few strong baselines due to the large number of models we evaluate. We make two notes on our human evaluation. First, we compared to **BlenderBot-3 which was trained with safety methods** (namely, fine-tuned on SaFeRDialogues). We argue that BlenderBot-3 serves as a sensible baseline for comparison to a safe and coherent dialogue system. Second, our main aim in conducting human evaluation was to show that response relevance is not degraded by using in-context learning.
>
> > *Since the proposed toxicity reduction method mainly relies on the prompts designed/retrieved by the authors, it should also be compared against other prompting based methods, especially since steering the behavior of LM using specialized prompts is not entirely new. See missing references for some related methods.*
>
> We thank Reviewer QjSk for their suggestion of including other prompting baselines in our work. We first highlight our inclusion of Self-Debias in our paper, a strong prompt-based detoxification procedure. Our method substantially outperformed Self-Debias (see Table 3). To further strengthen our paper, we have added two new prompting baselines which we describe below:
> 1. **Helpful and Harmless Prompting:** We prompt a model to be "helpful" and "harmless." For this baseline, we adopt a prompt from [Touvron et al., 2023](https://arxiv.org/abs/2307.09288) (LLaMA-2; see the fourth example in Table 39 in their work) for dialogue safety.
> 2. **Rule-of-Thumb Prompting:** We include rules-of-thumb/guidelines from ProsocialDialog in the prompt when performing response generation. To select the rule-of-thumb to include in-context, we take a rule-of-thumb from the top-ranked safety demonstration after retrieval.
>
> We provide an updated subset of results from Table 3 below. In general, we find the two new baselines outperform the base model (OPT-6.7B) but are outperformed by our method (OPT-6.7B + Dense). We hope these new baselines help to better contextualize our method amongst existing prompting strategies.
>
> | Model | Classifier ($\uparrow$) | Perspective ($\uparrow$) | Word List ($\uparrow$) |
> |-------|------------|-------------|-----------|
> | OPT-6.7B | 57.79 $\pm$ 0.79 | 74.35 $\pm$ 1.97 | 86.66 $\pm$ 2.04
> | OPT-6.7B + Dense **(ours)** | 77.57 $\pm$ 0.57 | 89.33 $\pm$ 0.09 | 94.22 $\pm$ 0.65
> | OPT-6.7B + Helpful/Harmless Prompt | 62.14 $\pm$ 0.05 | 80.92 $\pm$ 0.87 | 88.57 $\pm$ 0.01
> | OPT-6.7B + Rule-of-Thumb | 64.29 $\pm$ 1.06 | 83.53 $\pm$ 0.50 | 89.94 $\pm$ 0.85
>
> We will include these results and additional experimental details in the camera-ready version of the paper.
>
> > *GPT-3.5/4 based automatic evaluation may favor longer responses (Yizhong, et al, 2023). The paper mentioned that having 10 demonstrations may cause generated responses to be longer. How would this affect the results in Table 2 and 3?*
>
> Yes, our LLM-eval may exhibit a bias towards longer responses. We mention potential biases with LLM-eval on lines 513-515 of the paper and note these biases only impact Table 3. While we include an instruction in the prompt ("*do not allow the length of responses to influence your evaluation*", see Appendix F) to attempt to mitigate this bias, our results may still be impacted. We will further emphasize this limitation in the camera-ready version of the paper.
>
> > A few other prompting methods that relates/could be used to improve dialogue safety
> > * Si, C., Gan, Z., Yang, Z., Wang, S., Wang, J., Boyd-Graber, J.L., & Wang, L. (2022). Prompting GPT-3 To Be Reliable. ArXiv, abs/2210.09150.
> > * Madaan, A., Tandon, N., Gupta, P., Hallinan, S., Gao, L., Wiegreffe, S., Alon, U., Dziri, N., Prabhumoye, S., Yang, Y., Welleck, S., Majumder, B.P., Gupta, S., Yazdanbakhsh, A., & Clark, P. (2023). Self-Refine: Iterative Refinement with Self-Feedback. ArXiv, abs/2303.17651.
>
> We thank Reviewer QjSk for the additional prompting references. We will incorporate these into the camera-ready version of the paper!
>
> In general, we believe the modularity of our approach, alongside its strong performance (as highlighted by you and Reviewer kDkA), makes our paper exciting to the community at large. We argue that our comprehensive and detailed experiments (as noted by you and Reviewer kDkA), empirically demonstrate that our approach can be adopted for toxicity reduction in any deployed dialogue system. Furthermore, our method can easily be used in conjunction with other popular safety strategies, such as RLHF, making it practical for real-world use.
>
> Given our clarifications about human evaluation and our additional prompting baselines, would you kindly consider increasing your evaluation of the excitement of our work?

---

### Official Review · Reviewer_kDkA · 2023-08-04

**Soundness:** 4

**Excitement:**

3: Ambivalent: It has merits (e.g., it reports state-of-the-art results, the idea is nice), but there are key weaknesses (e.g., it describes incremental work), and it can significantly benefit from another round of revision. However, I won't object to accepting it if my co-reviewers champion it.

**Paper Topic And Main Contributions:**

This study addresses the subject of safe response generation, a prevalent topic within the field of NLP and dialogue systems.

The authors propose a retrieval and in-context learning-based method aimed at enhancing response safety. This involves retrieving safe demonstrations analogous to the context, followed by the application of in-context learning to augment dialogue safety. Through a series of both automatic and manual experiments, the paper provides evidence that (1) in-context learning can indeed enhance the safety of generated responses, and (2) the approach performs competitively with these baselines without requiring training and without degrading quality.

**Questions For The Authors:**

Aside from the aforementioned points of critique, I have another question.

What would be the consequences if the retrieval model were to retrieve an unsafe demonstration? Could this potentially make the conversation model more unsafe? It's important to note that it is possible that the retrieval database isn't entirely clean, and we shouldn't operate under the assumption that it is. The most hazardous scenario would be if the retrieval database were deliberately poisoned. How would your proposed method handle such situations?

**Reasons To Accept:**

1. Dialogue safety is an important topic and the results presented are encouraging and promising.
2. The authors have executed a comprehensive and robust set of comparative experiments to validate the efficacy of their proposed method. The evaluation metrics employed are diverse, adding to the credibility of the results.
3. The paper is well-written, easy to comprehend, and the structure is logically organized, making it an accessible read for individuals in the field.

**Reasons To Reject:**

1. The methodology proposed in this paper, while effective, is not particularly novel or innovative. It appears to be a standard approach in general dialogue response generation, and the paper predominantly focuses on verifying its effectiveness in the safety task. The lack of novelty reduces the impact of this study.
2. The choice of evaluation tool raises questions. The authors did not utilize the LLM-eval for safety evaluation, which could provide more accurate results than a safety classifier. The primary safety results (e.g., those presented in Figure 3, Table 1, and Table 3) should ideally be based on LLM-eval. Furthermore, tools like Perspective API and Word List are less effective in context-sensitive and implicit safety scenarios, which are more common in the LLM dialogue model era.

**Reproducibility:**

4: Could mostly reproduce the results, but there may be some variation because of sample variance or minor variations in their interpretation of the protocol or method.

**Reviewer Confidence:**

5: Positive that my evaluation is correct. I read the paper very carefully and I am very familiar with related work.

---

> ### Author Rebuttal · Authors · 2023-08-29
>
> We thank Reviewer kDkA for their clear feedback which will help improve our paper! We are glad they found our paper "**well-written,**" "**easy to comprehend,**" and "**logically organized.**" We are also pleased they found our results "**encouraging and promising**" and our experiments "**comprehensive and robust.**" We highlight that the other reviewers found:
> * Our paper topic "**timely**" (Reviewer yFp7).
> * Our experiments "**numerous**" and "**detailed**" in support of our claims (Reviewer QjSk).
> * The **competitive performance** of our approach relative to training-based safety methods compelling (Reviewer QjSk).
> * The **large-scale nature of our study of in-context learning for dialogue safety** compelling (Reviewer QjSk).
>
>
> We address each of Reviewer kDkA's concerns below.
>
> > *The methodology proposed in this paper, while effective, is not particularly novel or innovative. It appears to be a standard approach in general dialogue response generation, and the paper predominantly focuses on verifying its effectiveness in the safety task. The lack of novelty reduces the impact of this study.*
>
> To the best of our knowledge, our work is the first to investigate retrieving demonstrations for improving response generation. If Reviewer kDkA is aware of existing works which investigate retrieving demonstrations to improve dialogue response quality, we would greatly  appreciate knowing them so we can better position our paper among existing works.
>
> > *The choice of evaluation tool raises questions. The authors did not utilize the LLM-eval for safety evaluation, which could provide more accurate results than a safety classifier. The primary safety results (e.g., those presented in Figure 3, Table 1, and Table 3) should ideally be based on LLM-eval. Furthermore, tools like Perspective API and Word List are less effective in context-sensitive and implicit safety scenarios, which are more common in the LLM dialogue model era.*
>
> We thank Reviewer kDkA for raising this concern. **To clarify, our LLM-eval does indeed incorporate safety.** One of the criteria provided in the prompt for comparing responses is "respectfulness" (see Figure 13). Thus, we argue that the LLM-eval results presented in the paper do quantify response toxicity to some extent. If Reviewer kDkA feels strongly compelled, we can easily include LLM-based evaluation scores for *purely* safety in the camera-ready version of the paper.
>
> > *What would be the consequences if the retrieval model were to retrieve an unsafe demonstration? Could this potentially make the conversation model more unsafe? It's important to note that it is possible that the retrieval database isn't entirely clean, and we shouldn't operate under the assumption that it is. The most hazardous scenario would be if the retrieval database were deliberately poisoned. How would your proposed method handle such situations?*
>
> We thank Reviewer kDkA for raising this concern. Yes, our method may be susceptible to adversarial attacks on the retrieval database. We highlight the need for human verification of the safety demonstrations on lines 585-590 of the paper. We note that **our method is effective even with a limited pool of safety demonstrations.** For example, using a pool of 10 demonstrations gives ~90% of the performance of using a 42K demonstration pool on DiaSafety (see Figure 8). Since only a small number of demonstrations are required, this ensures minimal human verification of the safeness of demonstrations is needed. Furthermore, we also note that many other dialogue safety methods, such as safety filtering, are vulnerable to similar data poisoning attacks. An exciting strength of our method is the minimal human supervision required for strong performance!
>
> In summary, we believe an exciting draw of our work is in its modularity. Our approach can easily be used in conjunction with existing safety methods, such as RLHF and safety filtering. We believe this modularity, alongside the strong performance of our approach (as you and Reviewer QjSk highlighted), makes our work exciting to the community at large. Furthermore, the simplicity of our method will make it easily adoptable in real-world settings, increasing the impact of our work. Given your assessment and our clarifications about novelty and LLM-based safety evaluation, would you consider championing our paper for publication at this conference? We are happy to engage in further discussion.

---

### Official Review · Reviewer_yFp7 · 2023-08-05

**Typos Grammar Style And Presentation Improvements:** none
**Soundness:** 4

**Excitement:**

3: Ambivalent: It has merits (e.g., it reports state-of-the-art results, the idea is nice), but there are key weaknesses (e.g., it describes incremental work), and it can significantly benefit from another round of revision. However, I won't object to accepting it if my co-reviewers champion it.

**Missing References:**

not aware

**Paper Topic And Main Contributions:**

LLMs can generate toxic/biased/unsafe content. This is an unsolved problem today where lots of methods are being developed to address the challenges of this hard problem (guardrails, RLHF, training data, etc). This paper investigates a retrieval-based in-context learning approach for reducing bias and toxicity in responses from chatbots. The idea is to retrieve demonstrations of safe responses to similar dialogue contexts. The method performs competitively with existing approaches (safety classifiers/filters, safe response finetuning, RLHF, safe decoding with additional compute at inference) to dialogue safety. Also, reductions in toxicity obtained using this approach are not at the cost of response relevance based on the relevance measures tested.

**Questions For The Authors:**

- Is there any relevant comparisons that can be done with to Automatic prompt learning/generation methods (perhaps using the prosocial dialog and/or self-align and self-instruct type of methods)?
- have the authors looked at other toxicity classifiers and using more fine-grained classification (from AuditNLG, Huggingdace Evaluate, DiaSafety classifier, activefence API) to allow for in-depth analysis

**Reasons To Accept:**

- Very timely topic of study for LLM based research
- latest open ecosystem LLMs studied

**Reasons To Reject:**

- The paper seems limited in the scope of the study only looking at coarse grained toxicity mitigation using prompting with a retrieval dataset (many confounding variables of safety aren't well discussed that co-occur with toxicity (the instigator and yea-sayer framework seems very limited as well). Also, as mentioned in the limitations, there's no comparison with other promising ICL methods like social rules of thumb/guidelines based prompting)
- Toxicity is one of many real issue with LLMs today. Detoxifying LLMs requires a framework and a processing pipeline. Authors suggest the retrieval based demonstration approach as in-context learning based prompting using the ProSocial dialog dataset and show competitive performance compared to finetuning and other approaches. Analysis seems to be missing suggesting where this retrieval does not make sense, are there any topics where retrieval based approach might hurt relevance?
- Paper glosses over the problem of toxicity in a very general manner and does not provide enough analysis to support that demonstrations help with different types of toxicity classes and settings. It uses a response safety toxicity classifier (2 years old) for scoring the models - using other safety classifiers (eg. Diasafety dataset models, moral integrity corpus benchmark or recent classifiers used in the Salesforce AuditNLG and Huggingface Evaluate classifiers)
- It is not clear how robust is this method. Perhaps, using this approach, one can easily setup harmful retrieval based prompts to make the generations more toxic, hence the approach is easily hackable without offering guardrails for robustness.

**Reproducibility:**

4: Could mostly reproduce the results, but there may be some variation because of sample variance or minor variations in their interpretation of the protocol or method.

**Reviewer Confidence:**

4: Quite sure. I tried to check the important points carefully. It's unlikely, though conceivable, that I missed something that should affect my ratings.

---

> ### Author Rebuttal · Authors · 2023-08-29
>
> We thank Reviewer yFp7 for their feedback and constructive criticism. We are pleased they found our paper topic "**timely**" and appreciated our usage of "**open ecosystem LLMs**" within our work. We highlight that the other reviewers found:
> * Our results "**encourgaging and promising**" (Reviewer kDkA).
> * Our paper "**well-written,**" "**easy to comprehend,**" and "**logically structured**"" (Reviewer kDkA).
> * Our experiments "**numerous**" and "**detailed**" in support of our claims (Reviewer QjSk).
> * Our evaluation metrics "**diverse**" adding to the "**credibility**" of our results (Reviewer kDkA).
> * The **competitive performance** of our approach relative to training-based safety methods compelling (Reviewer QjSk).
> * Our experiments "**comprehensive and robust**" (Reviewer kDkA).
> * The **large-scale nature of our study of in-context learning for dialogue safety compelling** (Reviewer QjSk).
>
> We address each of Reviewer yFp7's concerns below.
>
> > *The paper seems limited in the scope of the study only looking at coarse grained toxicity mitigation using prompting with a retrieval dataset (many confounding variables of safety aren't well discussed that co-occur with toxicity (the instigator and yea-sayer framework seems very limited as well)*.
>
> We note that while the dialogue safety framework of [Dinan et al., 2021](https://arxiv.org/abs/2107.03451) may be limited, expanding upon and improving this framework is outside of the scope of our work. Furthermore, we note that this framework has been adopted in many recent works ([Dinan et al., 2022](https://aclanthology.org/2022.acl-long.284/); [Shuster et al., 2022](https://arxiv.org/abs/2208.03188)).
>
> > *Also, as mentioned in the limitations, there's no comparison with other promising ICL methods like social rules of thumb/guidelines based prompting)*
>
> We thank Reviewer yFp7 for this suggestion. To further strengthen our work, we have added two new prompting baselines. We describe each below.
> 1. **Helpful and Harmless Prompting:** We prompt a model to be "helpful" and "harmless." For this baseline, we adopt a prompt from [Touvron et al., 2023](https://arxiv.org/abs/2307.09288) (LLaMA-2; see the fourth example in Table 39 in their work) for dialogue safety.
> 2. **Rule-of-Thumb Prompting:** We include rules-of-thumb/guidelines from ProsocialDialog in the prompt when performing response generation. To select the rule-of-thumb to include in-context, we take a rule-of-thumb from the top-ranked safety demonstration after retrieval. We adapt the prompt from [Kim et al., 2022](https://aclanthology.org/2022.emnlp-main.267/) (see Appendix D in their work) for this baseline.
>
> We provide an updated subset of results from Table 3 below. In general, we find the two new baselines outperform the base model (OPT-6.7B) but are outperformed by our method (OPT-6.7B + Dense). We found these new baselines performed comparably to the base model in terms of response quality.
>
> | Model | Classifier ($\uparrow$) | Perspective ($\uparrow$) | Word List ($\uparrow$) |
> |-------|------------|-------------|-----------|
> | OPT-6.7B | 57.79 $\pm$ 0.79 | 74.35 $\pm$ 1.97 | 86.66 $\pm$ 2.04
> | OPT-6.7B + Dense **(ours)** | 77.57 $\pm$ 0.57 | 89.33 $\pm$ 0.09 | 94.22 $\pm$ 0.65
> | OPT-6.7B + Helpful/Harmless Prompt | 62.14 $\pm$ 0.05 | 80.92 $\pm$ 0.87 | 88.57 $\pm$ 0.01
> | OPT-6.7B + Rule-of-Thumb | 64.29 $\pm$ 1.06 | 83.53 $\pm$ 0.50 | 89.94 $\pm$ 0.85
>
> We hope these baselines better contextualize our method amongst existing prompting strategies. We will include these results and additional experimental details in the camera-ready version of the paper.
>
> > *Toxicity is one of many real issue with LLMs today. Detoxifying LLMs requires a framework and a processing pipeline. Authors suggest the retrieval based demonstration approach as in-context learning based prompting using the ProSocial dialog dataset and show competitive performance compared to finetuning and other approaches. Analysis seems to be missing suggesting where this retrieval does not make sense, are there any topics where retrieval based approach might hurt relevance?*
>
> Thank you to Reviewer yFp7 for this question. In Appendix B, we have already investigated the effectiveness of our approach with both *safe* and *unsafe* dialogue contexts in our work. For instance, with innocuous dialogue contexts, one might imagine that providing safety demonstrations may worsen response quality.
>
> We experimentally found that **providing safety demonstrations when they are not required does not worsen response quality.** For example, in Table 5 we present response relevance results for Commonsense-Dialogues. We observe that OPT-13B with zero in-context safety demonstrations obtains an F1 score of 11.01 while OPT-13B with ten in-context safety demonstrations obtains an F1 score of 11.61 (see lines 411-413). In general, we observed no substantial degradation of response relevance in *safe* or *unsafe* settings.
>
> > *Paper glosses over the problem of toxicity in a very general manner and does not provide enough analysis to support that demonstrations help with different types of toxicity classes and settings. It uses a response safety toxicity classifier (2 years old) for scoring the models - using other safety classifiers (eg. Diasafety dataset models, moral integrity corpus benchmark or recent classifiers used in the Salesforce AuditNLG and Huggingface Evaluate classifiers)*
>
> We selected these toxicity evaluation tools as they have seen widespread adoption in previous work. To demonstrate that our results are consistent across a range of toxicity classifiers, we have provided new results for a RoBERTa toxicity classifier trained on ToxiGen ([Hartvigsen et al., 2022](https://aclanthology.org/2022.acl-long.234/)) and a RoBERTa toxicity classifier trained using Dynabench ([Vidgen et al., 2021](https://aclanthology.org/2021.acl-long.132/); the default classifier used in Hugging Face Evaluate for toxicity). We provide results for DiaSafety below and report the percentage of safe responses for different models. We observe that **for all three classifiers, our method performs competitively with DIRECTOR.**
>
> | Model | Bot-Adversarial Dialogue Classifier ($\uparrow$) | Hugging Face Evaluate Toxicity ($\uparrow$) | ToxiGen ($\uparrow$) |
> |-|-|-|-|
> | OPT-6.7B | 57.79 $\pm$ 0.79 | 76.90 $\pm$ 0.90 | 60.95 $\pm$ 0.68 |
> | OPT-6.7B + Random | 68.31 $\pm$ 0.97 | 82.65 $\pm$ 0.32 | 64.54 $\pm$ 1.43|
> | OPT-6.7B + BM25 | 71.16 $\pm$ 0.92 | 83.59 $\pm$ 0.47 | 65.23 $\pm$ 1.32 |
> | OPT-6.7B + Dense | 77.57 $\pm$ 0.57 | 86.08 $\pm$ 0.88 | 69.19 $\pm$ 0.36 |
> | OPT-6.7B + DIRECTOR | 79.82 $\pm$ 1.15 | 85.29 $\pm$ 0.41 | 65.48 $\pm$ 0.42
>
> We will add these classifier results to the appendix of the paper.
>
> > *It is not clear how robust is this method. Perhaps, using this approach, one can easily setup harmful retrieval based prompts to make the generations more toxic, hence the approach is easily hackable without offering guardrails for robustness.*
>
> We thank Reviewer yFp7 for raising this. Yes, our method may be susceptible to adversarial attacks on the retrieval database. We highlight the need for human verification of the safety demonstrations on lines 585-590 of the paper. We note that **our method is effective even with a limited pool of safety demonstrations.** For example, using a pool of 10 demonstrations gives ~90% of the performance of using a 42K demonstration pool on DiaSafety (see Figure 8). Since only a small number of demonstrations are required, this ensures minimal human verification of the safeness of demonstrations is needed. Furthermore, we also note that many other dialogue safety methods, such as safety filtering, are vulnerable to similar data poisoning attacks. An exciting strength of our method is the minimal human supervision required for strong performance!
>
> > *Is there any relevant comparisons that can be done with to Automatic prompt learning/generation methods (perhaps using the prosocial dialog and/or self-align and self-instruct type of methods)?*
>
> We first highlight our inclusion of Self-Debias in the paper, a strong prompt-based detoxification procedure. Our method substantially outperformed Self-Debias (see Table 3). We also highlight the addition of our **Helpful and Harmless Prompting** and **Rule-of-Thumb Prompting** baselines which further help position our method amongst existing prompting strategies.
>
> > *have the authors looked at other toxicity classifiers and using more fine-grained classification (from AuditNLG, Huggingdace Evaluate, DiaSafety classifier, activefence API) to allow for in-depth analysis*
>
> If Reviewer yFp7 feels strongly compelled, we can easily add detailed Perspective API scores to the appendix of the camera-ready version of the paper.
>
> In general, we believe the **strong performance of our approach** (as noted by Reviewers kDkA and QjSk) alongside our **comprehensive experimental results** (as noted by Reviewers kDkA and QjSk), will make our paper exciting to the community. Furthermore, the modularity of our method will make it easily adoptable in practice and usable alongside other popular safety strategies, such as RLHF.
>
> In light of this discussion, could you reconsider your assessment of our work and consider increasing your scores? We are happy to engage further.

---

### Meta-Review · Area_Chair_fKsr · 2023-10-06

**Recommendation:** 4

**Metareview:**

The paper has details about comprehensive experiments, including investigations into model size and the order/number of safe demonstrations. Reviewers note that this first large-scale evaluation of in-context learning for dialogue safety, adding significant value to the field.

---

### Decision · Program_Chairs · 2023-10-07

**Decision:**

Accept-Findings

**Comment:**

The paper has details about comprehensive experiments, including investigations into model size and the order/number of safe demonstrations. Reviewers note that this first large-scale evaluation of in-context learning for dialogue safety, adding significant value to the field.